# Predictive Biomarkers for Outcomes of Immune Checkpoint Inhibitors (ICIs) in Melanoma: A Systematic Review

**DOI:** 10.3390/cancers13246366

**Published:** 2021-12-18

**Authors:** Joosje C. Baltussen, Marij J. P. Welters, Elizabeth M. E. Verdegaal, Ellen Kapiteijn, Anne M. R. Schrader, Marije Slingerland, Gerrit-Jan Liefers, Sjoerd H. van der Burg, Johanneke E. A. Portielje, Nienke A. de Glas

**Affiliations:** 1Department of Medical Oncology, Leiden University Medical Center, Albinusdreef 2, 2333 ZA Leiden, The Netherlands; J.C.Baltussen@lumc.nl (J.C.B.); M.J.P.Schoenmaekers-Welters@lumc.nl (M.J.P.W.); E.M.E.Verdegaal@lumc.nl (E.M.E.V.); H.W.Kapiteijn@lumc.nl (E.K.); M.Slingerland@lumc.nl (M.S.); S.H.van_der_Burg@lumc.nl (S.H.v.d.B.); J.E.A.Portielje@lumc.nl (J.E.A.P.); 2Oncode Institute, Leiden University Medical Center, Albinusdreef 2, 2333 ZA Leiden, The Netherlands; 3Department of Pathology, Leiden University Medical Center, Albinusdreef 2, 2333 ZA Leiden, The Netherlands; A.M.R.Schrader@lumc.nl; 4Department of Surgery, Leiden University Medical Center, Albinusdreef 2, 2333 ZA Leiden, The Netherlands; G.J.Liefers@lumc.nl

**Keywords:** melanoma, immune checkpoint inhibitor, prediction, response, biomarkers, systematic review

## Abstract

**Simple Summary:**

Immune checkpoint inhibitors (ICIs) have revolutionized treatment of advanced melanoma and survival of melanoma patients has radically improved since. However, as durable responses after ICIs are only observed in 30–50% of melanoma patients, there is an unmet need to identify predictive biomarkers for response. This systematic review demonstrates the substantial number of publications that have studied a wide variety of possible biomarkers. Covering 177 publications that investigated 128 unique biomarkers, we provide an overview of all studied biomarkers in correlation with response or survival. We highlight blood, tumor and fecal biomarkers that were associated with response to ICIs in multiple studies. Of these, only T-cell inflamed gene expression profiling was predictive for response in a large clinical trial and validated in other studies, thus representing a promising biomarker for clinical practice. Large validation studies are warranted to confirm the predictive utility of other biomarkers, thereby further personalizing immunotherapy treatment.

**Abstract:**

Immune checkpoint inhibitors (ICIs) have strongly improved the survival of melanoma patients. However, as durable response to ICIs are only seen in a minority, there is an unmet need to identify biomarkers that predict response. Therefore, we provide a systematic review that evaluates all biomarkers studied in association with outcomes of melanoma patients receiving ICIs. We searched Pubmed, COCHRANE Library, Embase, Emcare, and Web of Science for relevant articles that were published before June 2020 and studied blood, tumor, or fecal biomarkers that predicted response or survival in melanoma patients treated with ICIs. Of the 2536 identified reports, 177 were included in our review. Risk of bias was high in 40%, moderate in 50% and low in 10% of all studies. Biomarkers that correlated with response were myeloid-derived suppressor cells (MDSCs), circulating tumor cells (CTCs), CD8+ memory T-cells, T-cell receptor (TCR) diversity, tumor-infiltrating lymphocytes (TILs), gene expression profiling (GEP), and a favorable gut microbiome. This review shows that biomarkers for ICIs in melanoma patients are widely studied, but heterogeneity between studies is high, average sample sizes are low, and validation is often lacking. Future studies are needed to further investigate the predictive utility of some promising candidate biomarkers.

## 1. Introduction

Metastatic melanoma is a tumor with poor prognosis and the incidence will continue to rise in the years to come. In the Netherlands, the incidence of melanoma almost tripled in twenty years, from 2525 patients in 2000 to 6787 in 2020 [1]. However, since the introduction of novel therapeutic agents, such as immunotherapy with immune checkpoint inhibitors (ICIs) and targeted therapy, survival of patients with advanced melanoma has radically improved. Immune checkpoints, proteins expressed by T-cells, regulate T-cell functionality and act as gatekeepers of immune responses to prevent autoimmunity. Immune checkpoint inhibitors aim to unleash the compromised T-cells, thereby inducing an anti-tumor response [2].

The first ICI approved by the US Food and Drug Administration (FDA) in 2011 was ipilimumab, a cytotoxic-T-lymphocyte-associated-protein-4 (CTLA-4) targeting antibody. Anti-CTLA-4 antibodies block T-cell inhibition, thereby promoting T-cell activation and the anti-tumor response. Survival increased from 6.4 months to 10 months for melanoma patients treated with ipilimumab compared to a vaccine control or dacarbazine. Moreover, survival curves seemed to reach a plateau between two and three years, demonstrating that a proportion of patients experienced a sustained long-term survival. Yet, response rates were lower than 20% and immune-related adverse events grade 3–4 were seen in approximately 15–45% of all patients [3]. More promising results were seen in trials comparing ipilimumab to nivolumab or pembrolizumab, the latter which are anti-programmed death-1 (PD-1) antibodies. Anti-PD-1 antibodies aim to prevent inhibitory signaling in activated effector T-cells by blocking their binding to PD-L1/2, resulting in functional T-cells that are possibly able to kill tumor cells. Objective responses ranged from 20–30% for monotherapy to 40–50% for combination therapy of anti-CTLA-4 and anti-PD-1. Furthermore, grade 3–4 toxicity dropped to 11–16% for anti-PD-1 monotherapy. In combination therapy, however, grade 3–4 toxicity rates remain at 40% due to anti-CTLA-4 therapy [4,5,6].

Although ICIs, specifically anti-PD-1 therapy, show better response rates than previous therapies, durable response is still seen in a subset of patients and risk of severe toxicity, such as diarrhea, hepatotoxicity and skin rash, remains high, especially in combination treatment. Biomarkers are therefore needed to predict response to therapy, thereby maximizing the therapeutic benefit for potential responders and saving toxicity and high costs for patients that are unlikely to benefit from ICIs. Biomarkers associated with response to ICI therapy in melanoma patients range from lactate dehydrogenase (LDH) in peripheral blood to tumor mutation burden (TMB), PD-L1 immunohistochemical positivity, and tumor-infiltrating lymphocytes (TILs) in tumor tissue [7,8,9,10]. Apart from LDH, these markers have not yet been integrated into standard clinical care. Therefore, this systematic review aimed to provide a full overview of the evidence for blood, tumor and fecal biomarkers that correlate with response, progression-free survival, and overall survival for ICIs in metastatic melanoma patients.

## 2. Materials and Methods

### 2.1. Search Strategy

On the 18 June 2020, we searched in the following electronic bibliographic databases for our systematic review: PubMed, COCHRANE Library, Embase, Emcare, and Web of Science. The search strategy used a combination of the following terms: “melanoma”, “immune checkpoint inhibitors” OR “immunotherapy”, “response” OR “outcome”, “predictor” and “biomarkers” (complete search strategy in Appendix A). The search was performed together with a research librarian. Duplicates were removed in EndNote. Two investigators (J.B and N.G) independently screened titles and abstracts to determine eligibility. Together with J.P., the same investigators independently assessed the full-text articles of potentially relevant studies to verify if eligibility criteria were met. All full texts were appraised by two authors. Any disagreement was resolved by a third author.

### 2.2. Selection Criteria

We aimed to identify studies in advanced melanoma patients receiving ICIs that investigated the association between blood, tumor, and fecal biomarkers with outcomes. Included studies had to study either anti-CLTA-4 treatment (ipilimumab), anti-PD-1 (nivolumab, pembrolizumab,) or combination therapy. Included study designs were randomized clinical trials, cohort studies (both retrospective and prospective), and case-control studies. Baseline biomarkers as well as biomarkers during treatment were considered eligible. Outcomes of interest were response (both objective response rate (ORR) and clinical benefit), overall survival (OS), melanoma-specific survival (MSS), and progression-free survival (PFS). Systematic reviews, editorials, and case reports were excluded, as well as articles that were not written in English, animal studies, and reports studying the ICI tremelimumab (*N* = 1), as this therapy proved not to be effective [11]. We excluded publications that analyzed biomarkers in adjuvant chemotherapy, interleukins, vaccination treatment, or targeted therapy. Reports studying adverse events or pseudoprogression as a single outcome and reports studying imaging biomarkers were not eligible.

### 2.3. Data Extraction and Quality Assessment

This systematic review was reported following the Preferred Reporting Items for Systematic Reviews and Meta-analyses (PRISMA) reporting guidelines [12]. Study characteristics, including first author, journal, publication date, study design and phase, type and timing of biomarker, type of therapy, baseline patient characteristics such as type of melanoma, age, country, and follow-up, and outcomes were extracted. We reported both positive and negative study results. Due to heterogeneity of the included studies, a meta-analysis was not considered feasible.

We estimated risk of bias by using the QUIPS tool, a quality tool for prognostic factor studies that examines risk of bias by the following six domains: study participation, study attrition, prognostic factor measurement, outcome measurement, adjustment for other prognostic factors, and statistical analysis [13]. We classified risk of bias for adjustment as low if researchers adjusted outcomes for at least tumor load or LDH and metastasis for response and age or WHO status for survival, as these factors have shown to be important confounders. Selected thresholds and the rationale for these thresholds had to be mentioned to score a low risk of bias for prognostic factor measurement. Statistical analyses were assessed as high risk of bias if only *p*-values were mentioned, and a 95% confidence interval (CI) was not calculated. In case of disagreement between two authors, consensus was reached after discussion. We considered risk of bias high if three or more categories were reported as high risk [14]. When studies scored low in all categories or at least in the categories of attrition, outcome measurement and statistical reporting, the risk was defined as low. All other studies were scored as moderate risk of bias.

### 2.4. Data Presentation

Study characteristics and outcomes are presented in tables and separated into anti-CTLA-4, anti-PD-1, combination therapy and mixed therapy cohorts. Mixed cohorts included both patients receiving anti-CTLA-4, anti-PD-1 and combination therapy. Biomarkers that were investigated in more than one study are shown in graphs. The median number of included patients per biomarker with interquartile range (IQR) was calculated. Graphs were created with GraphPad PRISM 9.0.1 and BioRender.com (accessed on 24 November 2021) was used to create Figure 4.

## 3. Results

The initial literature search yielded 3940 publications, of which 1404 were duplicates. Of the unique 2536 publications, 1996 records were identified in PubMed, 389 in Web of Science, 21 in COCHRANE library and 130 in Embase. After title and abstract screening based on inclusion criteria as defined in the Methods section, 267 studies were eligible for our review. We extracted the full text of these publications and after screening thereof, we excluded 106 studies and 161 studies remained suitable for our review. Cross referencing identified another 16 relevant publications, which resulted in 175 original publications that we included in our review (Figure 1, Appendix A for complete reference list).

A description of the 177 identified reports is shown in supplemental Appendix A. Of all publications, 77% (*N* = 136) reported response as outcome, 66% (*N* = 117) studied OS and in 43% (*N* = 76) of all publications PFS was studied. Adjustment for confounders was performed in 69 articles. We found 24 reports that analyzed parameters in a development cohort as well as a validation cohort and 153 reports that only investigated biomarkers in a developmental study design. In most studies, 55–70% of the included patients were men. Information about race was only described in five studies.

In total, we identified 128 unique biomarkers. Most studies investigated biomarkers for anti-PD-1 therapy (*N* = 81) and anti-CTLA-4 therapy (*N* = 73), whereas 12 publications studied combination therapy (Table 1, Table 2, Table 3 and Table 4). Moreover, 24 publications were based on mixed cohorts with both anti-CTLA-4, anti-PD-1, and combination therapy (Table 4). Biomarkers were most frequently investigated before the start of treatment (*N* = 103), 71 reports analyzed them before and during treatment and three reports only measured markers during treatment. We assessed the quality and found that 40% of all publications (*N* = 70) were estimated high risk of bias, 50% (*N* = 89) were moderate risk of bias and 10% (*N* = 18) were estimated low risk of bias. Publications with low risk of bias mainly studied blood biomarkers.

### 3.1. Peripheral Blood Biomarkers

In total, we identified 72 unique blood biomarkers in 162 different publications (Figure 2, Appendix A). Soluble blood biomarkers that were frequently investigated were LDH, leukocyte counts including lymphocytes, neutrophils, eosinophils and the ratio between these cytology markers, myeloid-derived suppressor cells (MDSCs and subset monocytic MDSCs (moMDSCs)), natural killer (NK) cells, systemic inflammation markers such as cytokines and S100, and circulating tumor cells (CTCs). LDH was the most extensively studied marker in 5149 patients. In the majority of all studies a correlation with OS or PFS was found, but LDH was only associated with response in 9/23 reports. Four of these papers confirmed the association between LDH and OS or PFS in a separate validation cohort [15,16,17,18]. Weide and colleagues [18] investigated LDH in the largest cohort (*N* = 616) and demonstrated that a high LDH at baseline was associated with worse OS in both a discovery and confirmation cohort of melanoma patients treated with pembrolizumab.

Neutrophil-to-lymphocyte ratio (NLR) was investigated in 2605 patients and high NLR correlated with poor PFS in 9/10 studies and poor OS in 14/17 studies, whereas a correlation with response was found in 6/10 studies. The largest prospective cohort that studied NLR was the study of Ferrucci and colleagues [19]. In 720 melanoma patients treated with ipilimumab, they found that an elevated NLR at baseline was associated with both OS and PFS, but response was not investigated in this study. Only two studies validated the association between NLR and OS in a separate cohort [20,21].

Low MDSC frequencies were an indication for response and OS in all anti-CTLA-4 studies (*N* = 4) and anti-PD-1 studies (*N* = 1), with a moderate risk of bias in 4/5 studies [16,22,23,24]. MoMDSCs were the only blood parameter that inversely correlated with both response and survival in all anti-CTLA-4 cohorts (*N* = 4), but average risk of bias was high and sample sizes were relatively small (median patients per analysis = 39, interquartile range (IQR) 32–55) [25,26,27,28].

Various pro- and anti-inflammatory cytokines, such as interleukins and chemokines, were analyzed and yielded a few significant correlations for response. Risk of bias was high for the majority of the reports. We also found inconsistent results for S100B and S100A8/9, which are calcium-binding proteins that are increased in melanoma patients. NK cells were associated with response to anti-PD-1 antibodies in 3/4 studies, but not associated with response to anti-CTLA-4 antibodies. Risk of bias was high in 4/6 studies. CTCs or circulating tumor DNA were predominantly studied in mixed therapy cohorts (*N* = 3). Other publications investigated CTCs in anti-PD-1 therapy (*N* = 1), and combination therapy (*N* = 1). Four of these reports were based on prospective cohorts. All reports found an association between a decrease in CTCs and response, OS and PFS and 3/5 analyses were considered a moderate risk of bias [7,29,30,31,32].

Biomarkers that play a role in systemic T-cell regulation and activation, such as CD4+, CD45RA+ or CD8+ (effector) memory T cells, regulatory T cells (Tregs), PD-L1+ expression on T cells, T cell repertoire (TCR), and TIM3 or LAG-3 expression on T-cells were investigated in several reports. Increased CD8+ effector memory T cells, associated with long-lived anti-tumor immunity, positively correlated with response in 3/4 studies and with OS in 3/4 studies, but a median of 20 patients per analysis (IQR 13–36,5) were included which resulted in a high risk of bias in 4/5 studies [25,33,34,35,36]. A significant correlation between response and TCR diversity in peripheral blood was found in all anti-CLTA-4 (*N* = 2) and anti-PD-1 (*N* = 1) cohorts, but again sample sizes were too small to draw firm conclusions [37,38]. PD-1 expression on CD4 or CD8 cells was not associated with response in most (4/5) of the studies (median patients per study =IQR 30–113·5) [33,39,40,41,42]. We observed different results for serum Tregs, as they correlated with response in anti-CLTA-4 treated cohorts (*N* = 1) with melanoma patients, but not in anti-PD-1 treated melanoma patients (*N* = 1) [16,43,44].

### 3.2. Tumor Biomarkers

In the tumor tissue-based studies, we identified 55 different biomarkers in 78 publications (Figure 3, Appendix A). Tumor biomarkers that were studied included specific mutations (BRAF, NRAS, cKIT), differential expressed genes included in a gene expression profiling (GEP) score, TMB or neoantigen load (NAL), various T-cell regulation subsets (memory T-cells, regulatory T-cells, TILs), and other immune factors such as perforin or granzyme A or B. Immunohistochemical detected PD-L1 expression on tumor cells was widely studied in a total of 2416 patients. In the anti-PD-1 and mixed therapy cohorts, PD-L1 positivity on pre-treatment tumor cells was correlated with clinical benefit in 9/14 analyses, PFS in 2/5 analyses and OS in 4/8 analyses, although 64% of the studies were estimated as having a high risk of bias. In a large cohort of 405 melanoma patients who were treated with pembrolizumab, Daud et al. [10] showed that the highest response rates (53–57%) were found for PD-L1 positive tumors (corresponding to ≥10% staining), whereas patients with PD-L1 negative tumors (corresponding to <1% staining) showed response rates of 8–12%.

TMB and NAL, predominantly resulting from non-synonymous somatic mutations, might also play a role in response to ICIs. Conflicting results were found for TMB, as this biomarker was associated with response in 10/18 studies. More than half (61%) of these studies were assessed as high risk of bias and only two studies investigated TMB in a validation cohort [45,46]. Wood and colleagues investigated TMB in a large cohort (*N* = 302) with both anti-PD-1 and anti-CLTA-4 treated patients, and found no correlation between TMB and response [47]. However, TMB did correlate with response in a cohort of 150 melanoma patients treated with ipilimumab [48]. In anti-CTLA-4 and anti-PD-1 treated patients, responders had a higher NAL at baseline in 3/4 studies, but risk of bias was high in 3/4 studies [46,48,49,50].

We found 14 studies that calculated a T-cell inflamed GEP score, composed of different inflammatory genes related to T-cell surface markers, antigen presentation, chemokines, cytolytic activity, and adaptive immune resistance. T-cell inflamed GEP scores were associated with response in 7/9 of the anti-PD-1 cohorts and 4/4 of the anti-CTLA-4 cohorts, with high (42%) and moderate (58%) risk of bias reports [36,48,50,51,52,53,54,55,56,57,58,59,60,61]. Hamid and colleagues [53] investigated an 18-gene T-cell inflamed GEP, which consisted of IFN-responsive genes and was developed in a previous study with 81 melanoma patients treated with pembrolizumab [61] in a large phase Ib clinical trial (*N* = 655). Their findings revealed an association between GEP score and response to pembrolizumab in both treatment-naïve and treatment-exposed patients. A third study confirmed the association between the 18-gene GEP and response to pembrolizumab in a separate cohort (*N* = 89) [52]. Other studies investigated different immune-related genes in studies with smaller sample sizes.

Intratumor T-cell activation and regulation markers were evaluated in several reports. All studies (*N* = 6) investigating association between TILs and response in anti-CLTA-4, anti-PD-1 and mixed therapy cohorts found significant results and most studies (67%) were assessed as moderate risk of bias [9,50,62,63,64,65]. We found conflicting results for specific T-cell markers, such as CD3+, CD4+ and CD8+ T-cells.

### 3.3. Gut Microbiome

We found three prospective cohorts that investigated potential biomarkers in human stool of 170 melanoma patients in total. All three studies found an association between response and the gut microbiome and had a high or moderate risk of bias. Among responders, microbiomes were enriched with species such as Faecalibacterium and Bacteroidales [66,67,68]. Oral microbiomes were not associated with response (Appendix A).

## 4. Discussion

In this systematic review, we summarize predictive blood, tumor and fecal biomarkers for ICIs in melanoma patients (Figure 4). Our data show that an impressive number of studies have searched for potential biomarkers, but the average predictive quality is moderate, heterogeneity between studies is large, and only a few biomarkers were validated in a separate cohort.

LDH was the most extensively studied blood biomarker and reflects cancer metabolic activity. LDH was mainly associated with OS and PFS but less with response, suggesting that high LDH is a prognostic rather than predictive marker. Similarly, a high NLR predicts a poor prognosis moreso than a low response rate. Immune cells with suppressive functions, such as MDSCs, monocytic MDSCs or Tregs, were more often associated with response in ICI-treated patients. Higher frequencies of Tregs, which express FoxP3, were predominantly associated with response to anti-CTLA-4 therapy, but not to anti-PD-1 antibodies. This is consistent with the mechanism of action of anti-CTLA-4, as preclinical murine studies previously showed that Tregs represent direct target cells to anti-CTLA-4 therapy due to their CTLA-4 expression [69,70]. Furthermore, Martens and colleagues showed that low MDSCs were indicators of benefit for ipilimumab in a development and validation cohort, suggesting that patients with an immune response suppressed by means other than myeloid cells (e.g., Tregs) are more likely to respond to anti-CLTA-4 therapy [16].

TCR diversity in relation to ICIs has also been intensively studied, and several studies have shown that anti-CTLA-4 broadens the peripheral TCR repertoire, whereas anti-PD-1 expands some T-cell clones, which results in a skewed TCR repertoire [71,72]. A high TCR diversity with reduced clonal loss at baseline was reported in responding patients, but few patients were included in these studies. Similarly, CD8+ memory T-cells and CTCs might facilitate adequate patient selection for ICIs, but larger sample sizes are needed to test this hypothesis.

In tumor-tissue based publications, PD-L1 expression on pre-treatment tumor samples was the most frequently studied biomarker. In the majority of the anti-PD-1 and mixed therapy cohorts, PD-L1 expression on tumor cells correlated with response. However, PD-L1-negative patients also showed durable responses, suggesting that a negative finding is of limited significance, which justifies why PD-L1 has not been implemented as a biomarker for the treatment of melanoma. Moreover, interpretation of the publications was difficult due to different immunohistochemistry methods and definitions of positive staining. Inconsistent results were also found for the predictive value of TMB. For example, Johnson and colleagues [45] showed that responders had a high mutational load, but TMB only correlated with OS in a report by Hugo et al. [57]. Better studies with harmonized thresholds are needed to further assess both TMB and neoantigen load as predictors in melanoma patients. TILs and TIL subsets were frequently studied, and although TIL numbers were clearly associated with response to ICIs, inconclusive results were described for the TIL subsets. Therefore, further research of TILs is required as well.

An emerging concept is the analysis of the tumor immune microenvironment in the form of T-cell inflamed GEP. T-cell inflamed GEP in tumor tissue was predictive for response in most studies. Most GEP signatures are characterized by, among other things, the upregulation of IFN-γ signaling. As the IFN-γ signaling pathway can induce the expression of PD-L1 and PD-L2 on tumor cells and macrophages, this signature is critical for antitumor immunity. A promising thing about GEP is its ability to integrate the complex biology of multiple microenvironmental features in comparison with a single biomarker. As mentioned in the results section, an 18-gene T-cell inflamed GEP was associated with response to pembrolizumab in three studies, one of which was a large trial, indicating its suitability for clinical implementation. Indeed, robust and reproducible genomic-based platforms will be needed to implement T-cell inflamed GEP in daily oncology care.

Additionally, biomarkers in the intestinal microbiota are emerging to predict response to ICIs. Three studies showed that responders had an abundance of several bacterial species in the gut microbiome. Gopalakrishnan and colleagues therefore proposed that a “favorable” gut microbiome modulates anti-tumor immune responses due to increased antigen presentation and improved effector T-cell function [66]. Interestingly, several clinical trials are currently investigating the alteration of gut microbiota, e.g., with modified bacteria as adjuvant therapy for ICIs, implying that the microbiome might also be a possible target for cancer treatment.

We conducted an extensive literature search to identify predictive biomarkers, reported both positive and negative study results, and performed an adequate quality assessment. To our knowledge, this is the first formal systematic review since 2017 that updates current literature about both anti-CLTA-4, anti-PD-1 and combination therapy in melanoma patients. Compared to the review by Jessurun and colleagues [73], more studies that analyzed biomarkers for anti-PD-1 or combination therapy (*N* = 81) have emerged. Two other important strengths of our review are the large number of publications that we included and the formal bias assessment that we performed for all studies. Moreover, our tables summarizing outcomes per biomarker are a helpful tool for both clinical oncologists and researchers to check if a specific biomarker has been studied before. 

Of course, this review has its limitations. First, different thresholds, study methods and adjustment for confounders resulted in high heterogeneity. Second, our review comprises a few Japanese studies that included not only cutaneous melanoma patients but acral and uveal melanoma as well, thereby studying a heterogenous population. Outcomes in Japanese patients might thus be influenced due to different tumor types. Third, as mentioned before, validation of identified predictive markers was often lacking. Last, study results from four publications were too complex and too extensive to accurately summarize all outcomes in our tables [74,75,76,77].

Despite previous studies addressing the role of race, lifestyle habits and metabolic disorders as important confounders on the outcome of ICIs [78], none of the included publications adjusted results for these confounders. Future studies on predictive biomarkers should therefore take into account these confounders in their analyses. Regarding gender, male patients were slightly overrepresented in most studies. This overrepresentation might result from a higher incidence of melanoma in males.

While a substantial amount of developmental research on biomarkers for immunotherapy has been published in the past decade, validation studies are still limited. Instead of continuing the search for new biomarkers, future research should focus instead on validating existing biomarkers in large sample sizes.

## 5. Conclusions

Biomarkers for ICIs in melanoma patients are widely studied, and several biomarkers, such as (monocytic) MDSCs, TCR diversity, CTCs, TILs, T-cell inflamed GEP, and gut microbiomes, are associated with response to ICIs. Of these, only T-cell inflamed GEP was predictive for response in a large clinical trial and validated in two studies, representing a promising biomarker for clinical practice. Most studies carried a high or moderate risk of bias due to small sample sizes, no adjustment for confounders and no validation in a separate cohort. Therefore, large prospective studies with comparable thresholds that are adjusted for relevant confounders and validation are warranted to confirm their predictive utility and thereby further personalize immunotherapy treatment.

## Figures and Tables

**Figure 1 cancers-13-06366-f001:**
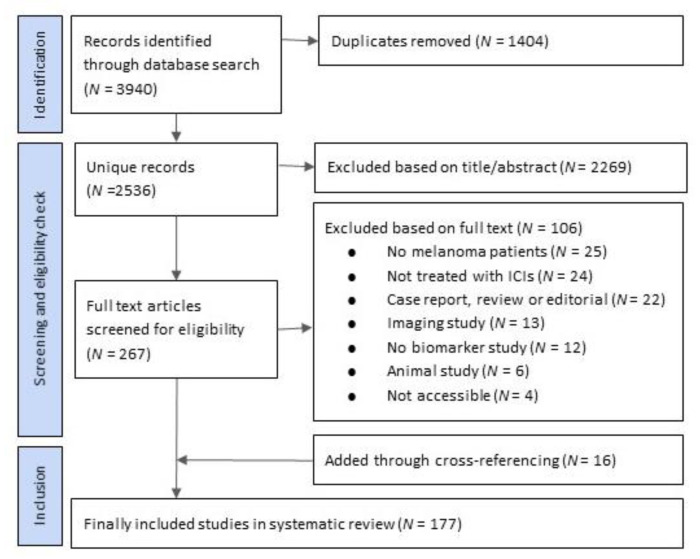
Study selection based on PRISMA methods.

**Figure 2 cancers-13-06366-f002:**
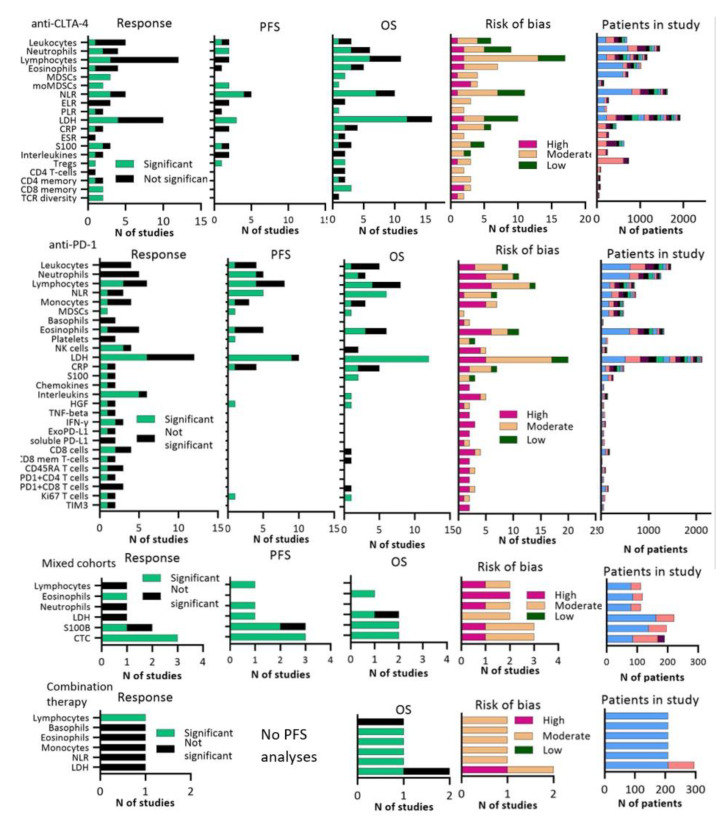
Graphs showing number of studies that were significantly associated or not associated to response, PFS, OS plus risk of bias per article for blood biomarkers. Right part of figure shows total number of included patients. A significant response was defined as *p* ≤ 0.05. Abbreviations; CRP:C-reactive Protein, CTC: circulating tumor cells, ELR: eosinophil-to-lymphocyte ratio, ESR: erythrocyte sedimentation ratio, HGF: hepatocyte growth factor, LDH: lactate dehydrogenase, NLR: neutrophil-to-lymphocyte ratio, PLR: platelet-to-lymphocyte ratio, Tregs: regulatory T-cells.

**Figure 3 cancers-13-06366-f003:**
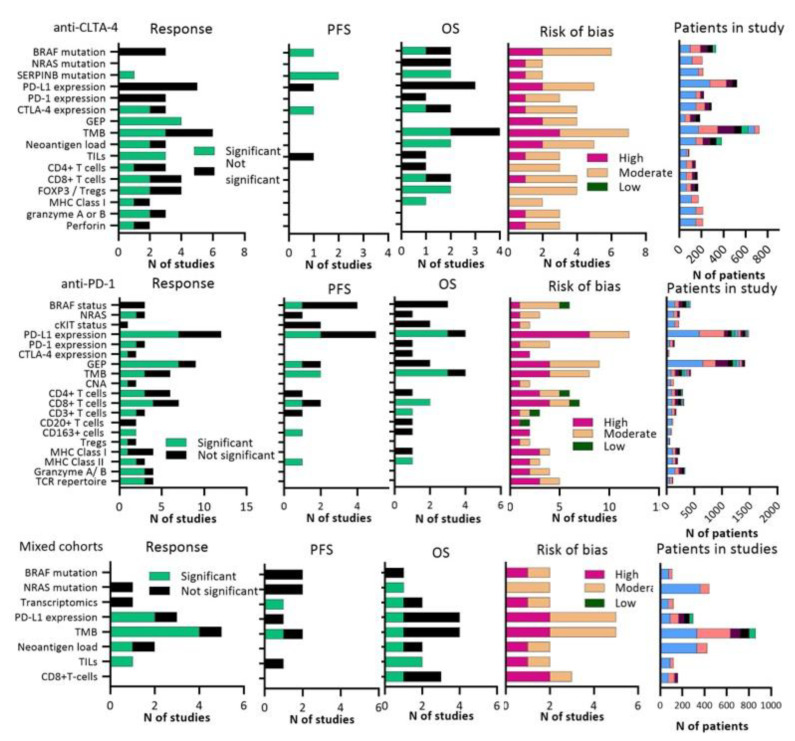
Graphs showing number of studies that were significantly associated or not associated to response, PFS, OS plus risk of bias per article for tumor tissue biomarkers. Right part of figure shows total number of included patients. A significant response was defined as *p* ≤ 0.05. Abbreviations; GEP: gene expression profiling, MHC: major histocompatibility complex, TILs: tumor-infiltrating lymphocytes, TCR: T-cell receptor, TMB: tumor mutation burden, Tregs: regulatory T-cells.

**Figure 4 cancers-13-06366-f004:**
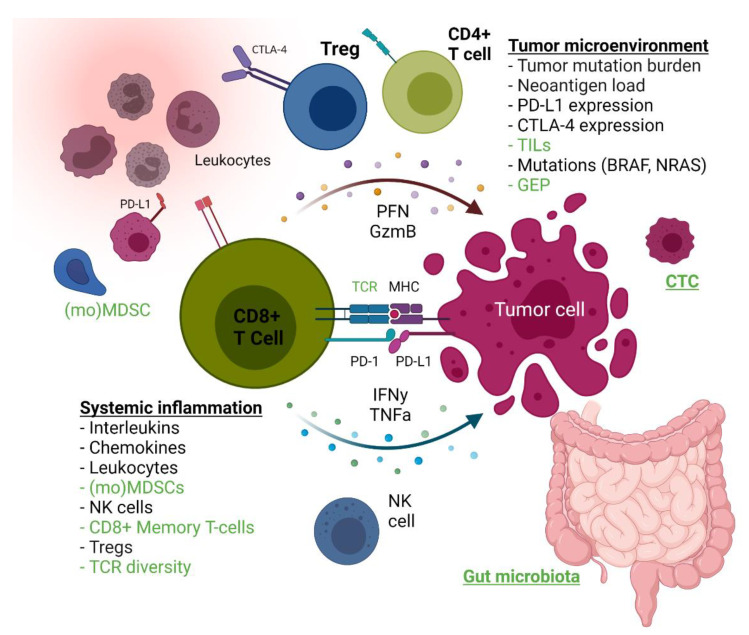
Overview of predictive biomarkers for response to ICIs. Biomarkers that were associated with response in most studies are marked green and biomarkers that were not associated with response in most studies are marked black. T cell infiltration markers, tumor cell microenvironment and gut microbiota. Abbreviations; CTC: circulating tumor cells, GzmB: Granzyme B, MDSCs: myeloid-derived suppressor cells, MHC: major histocompatibility complex, NK: natural killer, PFN: perforin, Tregs: regulatory T cells. Created with BioRender (BioRender.com, accessed on 24 November 2021).

**Table 1 cancers-13-06366-t001:** Summary of most important biomarkers that were studied for anti-CTLA-4 therapy and assessed risk of bias per article.

Biomarker	*N* of Studies	*N* of Patients	Median Patients per Article (IQR)	Response	PFS	OS	Quality Assessment
LDH	20	2539	86 (53–86)	LDH was associated with response in 4/10 studies, not associated in 6/10 studies.	LDH was associated with PFS in 3/3 studies.	LDH was associated with OS in 12/16 studies, not associated in 4/16 studies.	2/20 high risk, 14/20, moderate, 4/20 low risk of bias
NLR	11	1632	78 (43–184)	NLR was associated with response in 3/5 studies, not associated in 2/5 studies.	NLR was associated with PFS in 4/5 studies, not associated in 1/5 studies.	NLR was associated with OS in 7/10 studies, not associated in 3/10 studies	1/11 high risk, 6/11 moderate risk, 4/11 low risk of bias
TMB	7	724	64 (56–174)	TMB was associated with response in 3/6 studies, not associated in 3/6 studies.		TMB was associated with OS in 2/4 studies, not associated in 2/4 studies.	3/7 high risk, 4/7 moderate, 0/7 low risk of bias
Neoantigen load (NAL)	5	385	64 (54–107)	NAL was associated with response in 2/3 studies, not associated in 1/3 studies.		NAL was associated with OS in 2/2 studies.	2/5 high risk 3/5 moderate, 0/5 low risk of bias
PD-L1 expression on tumor cells	5	637	111 (48–214)	PD-L1 was not associated with response in 5/5 studies.	PD-L1 was not associated with PFS in 1/1 studies.	PDL-1 was not associated with OS in 3/3 studies.	2/5 high risk, 3/5 moderate, 0/5 low risk of bias
MDSCs	4	726	48 (22–475)	MDSCs were associated with response in 3/3 studies.		MDSCs were associated with OS in 2/2 studies.	1/4 high risk, 3/4 moderate risk, 0/3 low risk of bias
T-cell inflamed GEP	4	304	58 (33–192)	GEP was associated with response in 4/4 studies.			2/4 high risk, 2/4 moderate, 0/4 low risk of bias
Tregs in tumor tissue	4	169	38 (31–58)	Tregs were associated with response in 2/4 studies, not associated in 2/4 studies.		Tregs were associated with OS in 2/2 studies.	0/4 high risk, 4/4 moderate, 0/4 low risk of bias
monocytic MDSCs	4	168	39 (32–55)	moMDSCs were associated with response in 2/2 studies.	moMDSCs were associated with PFS in 2/2 studies	moMDSCs were associated with OS in 1/1 studies.	3/4 high risk 1/4 moderate risk, 0/4 low risk of bias
Tregs in blood	3	741	95 (31–615)	Tregs were associated with response in 1/1 studies.	Tregs were associated with RFS in 1/1 studies	Tregs were associated with OS in 2/2 studies.	1/3 high risk, 2/3 moderate risk, 0/3 as low risk of bias
CD8 memory T-cells in blood	3	90	30 (17–43)	CD8 memory T-cells were associated with response in 2/2 studies.		CD8 memory T-cells were associated with OS in 3/3 studies.	2/3 high risk, 1/3 moderate risk 0/1 low risk of bias
TILs	3	90	17 (9–64)	TILs were associated with response in 3/3 studies.	TILs were not associated with PFS in 1/1 studies.	TILs were not associated with OS in 1/1 studies.	1/3 high risk, 2/3 moderate, 0/3 low risk of bias
TCR diversity in blood	2	54	27 (N/A)	TCR diversity was associated with response in 2/2 studies.		TCR diversity was not associated with OS in 1/1 studies.	1/2 high risk, 1/2 moderate risk, 0/2 low risk of bias
NK cells in blood	2	63	32 (N/A)	NK cells were associated with response in 1/2 studies, not associated in 1/2 studies.			1/2 high risk, 1/2 moderate, risk 0/2 low risk of bias

Abbreviations: GEP: gene expression profiling, IQR: interquartile range, LDH: lactate dehydrogenase, MDSCs: myeloid-derived suppressor cells, NK: natural killer, NLR: neutrophil-to-lymphocyte ratio, TCR: T-cell receptor, TILs: tumor-infiltrating lymphocytes, TMB: tumor mutation burden, Tregs: regulatory T cells.

**Table 2 cancers-13-06366-t002:** Summary of most important biomarkers that were studied for anti-PD-1 therapy and assessed risk of bias per article.

Biomarker	*N* of Studies	*N* of Patients	Median Patients per Study (IQR)	Response	PFS	OS	Quality Assessment
LDH	20	2274	78 (39–152)	LDH was associated with response in 4/10 studies, not associated in 6/10 studies.	LDH was associated with PFS in 10/11 studies, not associated in 1/11 studies.	LDH was associated with OS in 13/13 studies.	5/20 high, 12/20 moderate, 3/20 low risk of bias
PD-L1 expression on tumor cells	12	1481	52 (30–68)	PD-L1 was associated with response in 7/12 studies, not associated in 5/12 studies.	PD-L1 was associated with PFS in 2/5 studies, not associated in 3/5 studies.	PD-L1 was associated with OS in 3/4 studies, not associated in 1/4 studies.	8/12 high, 4/12 moderate, 0/12 low risk of bias
T-cell inflamed GEP	9	1237	58 (33–192)	GEP was associated with response in 7/9 studies, not associated in 2/9 studies.	GEP was associated with PFS in 1/2 studies, not associated in 1/2 studies	GEP was not associated with OS in 2/2 studies.	4/9 high, 5/9 moderate, 0/9 low risk of bias
NLR	8	732	77 (41–138)	NLR was associated with response 1/3 studies, not associated in 2/3 studies.	NLR was associated with PFS in 5/5 studies.	NLR was associated with OS in 6/6 studies.	1/8 high, 6/8 moderate, 1/8 low risk of bias
TMB	8	68	52 (41–67)	TMB was associated with response in 3/6 studies, not associated in 3/6 studies.	TMB was associated with PFS in 2/2 studies.	TMB was associated with OS in 3/4 studies, not associated in 1/4 studies.	4/8 high, 4/8 moderate, 0/8 low risk of bias
NK cells in blood	5	128	20 (13–41)	NK cells were associated with response in 3/4 studies, not associated in 1/4 studies.		NK cells were not associated with OS in 2/2 studies.	4/5 high, 1/5 moderate, 0/5 low risk of bias
TCR diversity in tumor	4	184	52 (22–57)	TCR diversity was associated with response in 3/4 studies, not associated in 1/4 studies.			2/4 high, 2/4 moderate, 0/4 low risk of bias
Gut microbiomes	2	104	52 (N/A)	Gut microbiomes were associated with response in 2/2 studies	Gut microbiomes were associated with PFS in 1/1 studies		0/2 high, 2/2 moderate, 0/2 low risk
CD8 memory T-cells in blood	2	29	15 (N/A)	CD8 memory cells were associated with response in 1/2 studies, not associated in 1/2 studies.		CD8 memory cells were not associated with OS in 1/1 studies.	2/2 high, 0/2 moderate, 0/2 low risk of bias
TILs	2	121	60 (N/A)	TILs were associated with response in 2/2 studies			2/2 high, 0/2 moderate, 0/2 low risk of bias
ctDNA	1	85	N/A		ctDNA was associated with PFS in 1/1 studies.	ctDNA was associated with OS in 1/1 studies.	0/1 high, 1/1 moderate, 0/1 low risk of bias
MDSCs	1	92	N/A	MDSCs were associated with response in 1/1 studies.	MDSCs were associated with PFS in 1/1 studies.	MDSCs were associated with OS in 1/1 studies.	0/1 high, 1/1 moderate, 0/1 low risk of bias
Tregs in blood	1	46	N/A	Tregs were not associated with response in 1/1 studies.			1/1 high, 0/1 moderate, 0/1 low risk of bias
TCR diversity in blood	1	38	N/A	TCR diversity was associated with response in 1/1 studies.			0/1 high, 1/1 moderate, 0/1 low risk of bias

Abbreviations: ctDNA: circulating tumor DNA, GEP: gene expression profiling, IQR: interquartile range, LDH: lactate dehydrogenase, MDSCs: myeloid-derived suppressor cells, NK: natural killer, NLR: neutrophil-to-lymphocyte ratio, TCR: T-cell receptor, TILs: tumor-infiltrating lymphocytes, TMB: tumor mutation burden, Tregs: regulatory T cells.

**Table 3 cancers-13-06366-t003:** Summary of most important biomarkers that were studied for combination therapy and assessed risk of bias per article.

Biomarker	N of Studies	N of Patients	Median Patients per Study (IQR)	Response	PFS	OS	Quality Assessment
LDH	2	295	148	LDH was associated with response in 1/2 studies, not associated in 1/2 studies.	LDH was associated with PFS in 1/1 studies.	LDH was associated with OS in 2/2 studies.	0/2 high risk, 1/2 moderate, 1/2 low risk of bias
NLR	1	209	N/A	NLR was not associated with response in 1/1 studies.		NLR was associated with OS in 1/1 studies.	0/1 high risk, 1/1 moderate 0/1 low risk of bias
TCR diversity	1	80	N/A		TCR diversity was associated with PFS in 1/1 studies.		0/1 high risk 1/1 moderate, 0/1 low risk of bias
Memory T-cells in tumor tissue	1	57	N/A		Memory T-cells were associated with PFS in 1/1 studies.		0/1 high risk 1/1 moderate, 0/1 low risk of bias
T-cell inflamed GEP	1	57	N/A	GEP was associated with response in 1/1 studies.			0/1 high risk 1/1 moderate, 0/1 low risk of bias
ctDNA	1	35	N/A	ctDNA was associated with response in 1/1 studies.	ctDNA was associated with PFS in 1/1 studies.	ctDNA was associated with OS in 1/1 studies.	1/1 high risk, 0/1 moderate 0/1 low risk
TMB	1	35	N/A	TMB was associated with response in 1/1 studies.		TMB was not associated with OS in 1/1 studies.	1/1 high risk, 0/1 moderate, 0/1 low risk of bias

Abbreviations: ctDNA: circulating tumor DNA, GEP: gene expression profiling, IQR: interquartile range, LDH: lactate dehydrogenase, NLR: neutrophil-to-lymphocyte ratio, TCR: T cell receptor, TMB: tumor mutation burden.

**Table 4 cancers-13-06366-t004:** Summary of most important biomarkers that were studied for mixed cohorts and assessed risk of bias per article.

Biomarker	N of Studies	N of Patients	Median Patients per Study (IQR)	Response	PFS	OS	Quality Assessment
TMB	5	861	91 (68–317)	TMB was associated with response in 4/5 studies, not associated in 1/5 studies.	TMB was associated with PFS in 1/2 studies, not associated in 1/2 studies.	TMB was associated with OS in 1/4 studies, not associated in 3/4 studies.	2/5 high risk, 3/5 moderate, 0/5 low risk of bias
PD-L1 expression on tumor cells	5	298	51 (1–84)	PD-L1 expression was associated with response in 2/3 studies, not associated in 1/3 studies.	PD-L1 expression was not associated with PFS in 1/1 studies.	PD-L1 was associated with OS in 1/4 studies, not associated with OS in 3/4 studies.	2/5 high risk, 3/5 moderate, 0/5 low risk of bias
Circulating tumor cells	3	190	82 (22–86)	CTCs were associated with response in 3/3 studies.	CTCs were associated with PFS in 3/3 studies.	CTCs were associated with OS in 2/2 studies.	1/3 high risk, 2/3 moderate 0/3 low risk
LDH	2	141	71 (N/A)	LDH was not associated with response in 1/1 studies.	LDH was associated with PFS in 1/2 studies.	LDH was associated with OS in 1-2 studies, not associated with OS in 1/2 studies.	0/2 high risk, 2/2 moderate, 0/2 low risk of bias
Neoantigen load (NAL)	2	423	212 (N/A)	NAL was associated with response in 1/2 studies, not associated in 1/2 studies.		NAL was associated with OS in 1/2 studies, not associated with OS in 1/2 studies.	2/2 high risk, 0/2 moderate, 0/2 low risk of bias
TILs in tumor tissue	2	123	62 (N/A)	TILs were associated with response in 1/1 studies.	TILs were not associated with PFS in 1/1 studies.	TILs were associated with OS in 2/2 studies.	1/2 high risk 1/2 moderate, 0/1 low risk of bias
Gut microbiomes	2	66	33 (N/A)	Gut microbiomes were associated with response in 2/2 studies.			0/2 high risk, 2/2 moderate, 0/2 low risk
NLR	1	32	N/A	NLR was not associated with response in 1/1 studies.			1/1 high risk, 0/1 moderate 0/1 low risk of bias
Tregs in tumor tissue	1	32	N/A			Tregs were not associated with OS in 1/1 studies.	1/1 high risk, 0/1 moderate 0/1 low risk of bias

Abbreviations: IQR: interquartile range, LDH: lactate dehydrogenase, NLR: neutrophil-to-lymphocyte ratio, TILs: tumor-infiltrating lymphocytes, TMB: tumor mutation burden, Tregs: regulatory T cells.

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
