# Peer review of "Predictive Biomarkers for Outcomes of Immune Checkpoint Inhibitors (ICIs) in Melanoma: A Systematic Review"

_cancers, 2021, doi:10.3390/cancers13246366_

Round 1

Reviewer 1 Report

This is a comprehensive systematic review of several studies investigating predictive and prognostic biomarkers of ICI therapy in melanoma. The number of included studies is very high. The investigators did a thorough analysis of the most relevant studies about peripheral blood, tissue, and intestinal microbiome potential biomarkers for ICI in metastatic melanoma.

I do not have any major criticisms or suggestions for this study, I think it is very clear and although deals with impressive amount of data, the analysis and conclusions are straightforward.

I found a couple of very minor issues:

  1. In Supplemental Table 1, the reference by Chen et al (Cancer Discovery) states "mucoasol melanoma", should be "mucosal melanoma"
  2. References 74 and 75 are missing in the submitted version of the manuscript.

Author Response

This is a comprehensive systematic review of several studies investigating predictive and prognostic biomarkers of ICI therapy in melanoma. The number of included studies is very high. The investigators did a thorough analysis of the most relevant studies about peripheral blood, tissue, and intestinal microbiome potential biomarkers for ICI in metastatic melanoma.

I do not have any major criticisms or suggestions for this study, I think it is very clear and although deals with impressive amount of data, the analysis and conclusions are straightforward.

We thank the reviewer for the kind words.

I found a couple of very minor issues:

Point 1: In Supplemental Table 1, the reference by Chen et al (Cancer Discovery) states "mucoasol melanoma", should be "mucosal melanoma"

Response 1: We agreed with the reviewer and changed the word “mucoasol” to “mucosal” in the Appendix.

Point 2: References 74 and 75 are missing in the submitted version of the manuscript.

Response 2: We agreed with the reviewer and added the missing references.

Reviewer 2 Report

In the present systematic review, authors screen the large set of published evidence on immunotherapy treatment in melanoma patients and shortlist 128 biomarkers (blood, fecal, and tumor) correlated with survival and immunotherapy responsiveness. Among the biomarkers, T‐cell inflamed gene expression profiling was of predictive value in trials. I have several reservations, my comments are appended as below:

  1. As immune checkpoints are the central theme, authors should first provide their basis and components in the introduction section. Authors may refer: PMID: 33076303, PMID: 34572799.
  2. Line 70- elaborate on the toxicities observed.
  3. Table 1-3- authors should also include statistical inference.
  4. As the study involves a vast number of cohorts possibly including patients with diverse races and ethnicity, do authors think race and ethnicity may influence the selection of biomarkers?
  5. Among the peripheral blood-based markers, are there any studies involving circulating tumor cells or metabolic products as glutamate?
  6. In studies with peripheral blood-based markers, do authors observe any gender bias?
  7. Among the tumor biomarkers, was PD-L1 only membrane-bound or also includes soluble form?
  8. Along with the studies markers, are there any pieces of evidence on other cofounders as lifestyle habits and metabolic disorders (PMID: 33076303)? Authors should include a note in the discussion part.
  9. The author should include a future directions segment.

Author Response

Please see the attachment for the point-by-point response to the reviewer´s comments in the cover letter. 

Round 2

Reviewer 2 Report

I congratulate the authors for the revised manuscript, with that the manuscript is closer to publication. I however suggest taking note of the following minor points:

  1. Figures 2, 3- define the significant response in the figure legend.
  2. Line 239- correct.
  3. line 220- note the type of cancer.

Author Response

Please see attachment for the Cover Letter.

Reviewer #2:

I congratulate the authors for the revised manuscript, with that the manuscript is closer to publication.

I however suggest taking note of the following minor points:

Point 1: Figures 2, 3- define the significant response in the figure legend.

Response 1: We agree with the reviewer and added this information in the figure legends from Figures 2 and 3.

“A significant response was defined as p ≤ 0.05.”

Point 2: Line 239- correct.

Response 2: We changed “Daud et [10] al” in “ Daud et al [10]” in Line 239.

In a large cohort of 405 melanoma patients who were treated with pembrolizumab, Daud et al [10] showed that the highest response rates (53-57%) were found for PD-L1 positive tumors (corresponding to ≥ 10% staining), whereas patients with PD-L1 negative tumors (corresponding to < 1% staining) showed response rates of 8-12%.

Point 3: Line 220- note the type of cancer.

Response 3: We agree with the reviewer and added the type of cancer.

We observed different results for serum Tregs, as they correlated with response in anti-CLTA-4 treated cohorts (N=1) with melanoma patients, but not in anti-PD-1 treated melanoma patients (N=1) [16, 43, 44].”

Round 3

Reviewer 2 Report

All my concerns are now addressed.